# Earth Sciences Teaching Difficulties in Secondary School: A Teacher's Point of View

**Radouan Chakour [1], Anouar Alami [1,\*], Sabah Selmaoui [1,2], Aâtika Eddif [3], Moncef Zaki [1] and Youssef Boughanmi [4]**

1. LIRDIST, Interdisciplinary Laboratory of Research in Didactics of Sciences and Technology, Faculty of Sciences Dhar Mahraz, Sidi Mohammed Ben Abdellah University, BP 2626, Fes 30000, Morocco; radouan.chakour@usmba.ac.ma (R.C.); sselmaoui@gmail.com (S.S.); zaki.moncef@yahoo.fr (M.Z.)
2. EREF, Team Education Research and Training, ENS, Cadi Ayyad University, Marrakech 40130, Morocco
3. Laboratory of Scientific Research and Pedagogical Development, Regional Center for Education and Training, Region of Fes-Meknes, Meknes B.P. 50010, Morocco; Aeddif@yahoo.fr
4. Georges Chevrier Center (CGC), University of Burgundy, BP 21000 Dijon, France; youssef.boughanmi@u-bourgogne.fr
* Correspondence: anouar.alami@usmba.ac.ma; Tel.: +212-661-796-480

**Abstract:** The teaching of Earth Sciences (ES) is particularly delicate and seems to be problematic for both learners and Moroccan teachers for multiple reasons. Based on this observation, this study aims at identifying the difficulties related to the teaching of ES by exploring the points of view of the Moroccan teachers toward this field. As an investigative tool, we used a questionnaire and semi-directive interviews with nearly 122 secondary school teachers of Life and Earth Sciences (LES). The results of our survey revealed that the major difficulties that hinder the teaching of natural sciences are mainly related to the teachers' university studies. Most of them had training in biology as well as in the relationship that the natural sciences maintain within time and space, the limited abstraction capacity of unmotivated learners, and the inadequacy of their prerequisites in these sciences. On the other hand, they were aware of the demotivating geological knowledge taught to the learners and the lack of initial and continuous training for teachers, especially for those who specialized in natural sciences.

**Keywords:** teachers; earth sciences; difficulties; secondary

## 1. Introduction

Earth Sciences (ES) is a branch of natural sciences such as biology (zoology, botany . . . ). ES are diachronic sciences relating to phenomena that take place over time [1]. They aim to define both the present functioning of the planet and the reconstruction of its history, which is spread across space and time. The notions of space and time on which the natural sciences are based go beyond the learners' individual experience. Geology is considered both a historical and a functionalist science [2] functionalist because it studies current external (erosion, transport, sedimentation) and internal geological phenomena (magmatism, metamorphism, earthquakes), and historical because it reconstructs the Earth's past. This past is interpreted from the recordings of geological phenomena in geological formations. Among the scientific discoveries that have revolutionized the status of the human being lies the discovery of "deep time" as its existence for millennia represents mere milliseconds on the time scale. This revolution has solved certain enigmas in other disciplines such as cosmology and biology. When asked about the process, Le Pichon (2003) stated that it corresponds to "all that makes the Earth not the same at this moment as it was a minute or 100,000 years ago or

1 million years" [3]. However, learners who are willing to learn and explain these geological processes need a cognitive basis to understand the geological time scale.

The teaching and learning of ES pose enormous difficulties, and the study of these difficulties can be applied both for the learners and for the Moroccan teachers for multiple reasons. Moreover, it can also be applied at the secondary and university levels; it was also the subject of various didactic research. Indeed, several didactics have focused on the understanding of time by learners [4], the origin of the Earth [5], the origin of life [6], and biological crises [7]. This work has shown that learners' difficulties in acquiring ES knowledge are largely related to the relationship of this discipline with time. As Gohau stated [8], "one of the difficulties of teaching ES is our inability to recognize the immense durations of the Earth's past [...]. We are as destitute in front of these enormous chronologies as before the immensity of space in astronomy."

Concerning the students' perceptions of ES, the work of Bezzi [9] showed that ES are perceived as approximate and subjective by geosciences students because they rely more on historical or narrative explanations, and they exclude all experimental approaches. Moreover, they oppose geosciences and considerphysical sciences to be objective and rigorous because the latter relies on theories that can be experimentally confirmed. The theory of evolution, for example, is seen as an important theme that brings to the school a broader perspective of natural phenomena and the nature of science [10].

On the other hand, the educational technology research team (2004) showed that the lack of fieldwork and experimental practice in the classroom as well as the lack of initial or continuous training of natural science professors in geology are the main difficulties that influence the teaching of this discipline. In the Lebanese context, Chalak and El Hage [11] have both argued that teaching ES encounters several epistemological, didactic, ideological, economic, curricular, and professional barriers. As for Morocco, there is little didactic work devoted to the teaching and learning of geosciences compared to those of physics and biology.

According to our professional and formative experience, it turns out that learners are not very interested in this discipline in high school, and they rarely choose professions related to this field (e.g., geophysics, hydrogeology, petrology, oceanography). However, it holds a significant share of the job market due to the fact that Morocco is rich with natural resources. The results of a study conducted by our research team [12] have shown that the ES discipline is marginalized because it is not taught at all scientific levels. It is taught in the first and second year of junior high school (with students 13 to 15 years old) and in the second and third year of high school (16 to 18 years old), but it was removed from primary school syllabi since the 1999 reform. Thus, it represents only 18% of the certificate-based evaluations in the current baccalaureate of scientific classes and the natural science stream.

Teachers, especially those specializing in biology, according to their pointofview, have experienced disaffection during their university studies and professional training, and they face great difficulties in teaching ES. They consider teaching this discipline one of the most difficult tasks [13], and since the teacher is a principal actor of the didactic transposition, he is able to influence this transposition by his opinions, his values, and his personal and professional practices [14]. In the present research, we are interested in studying the difficulties encountered by teachers of ES. We aim to do so by exploring their opinions related to teaching ES in order to identify the origin of these difficulties. We pose the following questions:

1. What are the views of secondary school teachers on teaching ES?
2. What are the difficulties faced by LES teachers in teaching and learning ES?

## 2. Materials and Methods

As a first step, we administered an anonymous questionnaire (see Supplementary S1) to secondary school teachers of Life and Earth Sciences (LES). As a second step, we conducted semistructured interviews to collect as much information as possible (see Supplementary S2).

Our sample consisted of Moroccan teachers of secondary school LES spread across several secondary schools.

A questionnaire was administered and put online via Google Forms in order to reach many different schools. We asked the teachers to complete the questionnaire via email through groups of LES teachers on social networks (mainly on Facebook). Thenumber of responses received was 150, and we had a return of 122 completed questionnaires.

Our sample was consisting of secondary teachers whose seniority, professional training and specialty are varied:

### 2.1. Seniority

The sample in our survey was very heterogeneous, composed mainly of teachers with less than 10 years of seniority (53.5 + 21.1 = 74.6%), 53.5% with less than 5 years, and 21.1% between 6 and 10 years. This is a young population.

### 2.2. Professional Training

The surveyed teachers had varied academic and professional training; 44.4% had a BA in biology or geology, 40.2% teachers had a Master's degree or a DESS (diploma of specialized graduate studies), 9.8% had a Ph.D., and only 4.1% had a Bac +1 or +2 (alumni as admission to secondary school teacher training entering exam was done with the Bac).

### 2.3. Specialty

Concerning the specialty of the subjects in our sample, most of the teachers (73.7%) specialized in life sciences, and only 18.4% had pursued university studies in ES. The specialty is an important variable in relation to our problem, as we think that teachers who specialize in biology might have more difficulties in teaching the necessary knowledge related to ES compared to those who specialized in ES.

These teachers are spread across many institutions belonging to the regional academies for training and education in Morocco. The online availability of our questionnaire played a key role in having this excellent representation of the target population in our sample. Table 1 summarizes the characteristics of our sample.

**Table 1.** Characteristics of the sample.

| | | |
|---|---|---|
| | ≥16 years | 8% |
| Seniority | From 11 years to 15 years | 18% |
| | From 6 years to 10 years | 21% |
| | From 1 year to 5 years | 53% |
| | Baccalaureate + 1 or 2 years in higher studies | 4% |
| Level of education | Baccalaureate + 3 or 4 years in higher studies | 44% |
| | Baccalaureate + 5 or 6 years in higher studies | 40% |
| | Baccalaureate + 8 years | 10% |
| | Other | 2% |
| | Biology | 74% |
| Specialty | Geology | 18% |
| | Other | 8% |

We delivered the questionnaire to identify the difficulties that LES teachers encounter in teaching ES and their origins. Some of the items had already been established by experts, while others were developed by us. Our questionnaire was validated by Moroccan, French, and Tunisian educational

experts (11 experts). These experts verified scientific and grammatical validity of the questionnaire. Their comments have been blessed to improve the structure of our tools

The questionnaire was introduced by explaining the purpose and goals of the study. Participants were asked to participate and were informed of their guaranteed anonymity. Participants were also informed that they could refuse to participate in the survey, and those who gave their consent were invited to complete the questionnaire.

We prepared and administered 6 questions, which consisted ofone open question (Q2) and 5 closed questions with multiple choice (MCQ) (Q1, Q3, Q4, Q5, Q6), whereby respondents should choose one or more answers.

The questionnaire contained six items whose objectives appear in the following Table (Table 2).

**Table 2.** Objectives of the questionnaire items.

| Questions | Objectives |
|---|---|
| Question 1. Closed question with multiple choice (MCQ) | Teaching difficulties of the earth sciences that teachers face. |
| Question 2. Open question | The underlying causes of the difficulties faced by teachers in teaching ES. |
| Question 3. Closed question (MCQ) | Teaching materials/tools used by teachers to transpose geological knowledge and knowledge. |
| Question 4. Closed question(MCQ) | What is the usefulness of ES learning for a learner? |
| Question 5. Closed question(MCQ) with a Likert scale | Are LES teachers familiar with the following fields in ES? |
| Question 6. Closed question (MCQ) | What is the reaction of teachers if learners interrupt them in class to tell them that learning ES is useless for them? |

ES: Earth Sciences, LES: Life and Earth Sciences.

The statistical analysis of the responses was performed using IBM SPSS (Statistical Package for the Social Sciences) 22. The percentages that appear in the tables are expressed according to the number of people questioned.

To complete the information collected by the questionnaire, we conducted semistructured interviews with 15 subjects from our sample, chosen in a random manner. The interviews took place in the educational institution's subjects in a friendly atmosphere and we have used the content to analyze the data obtained, which we have categorized to facilitate the analysis.

## 3. Results

### 3.1. The Difficulties of Teaching ES in Morocco

To regard difficulties of teaching ES we have collected the following data represented in Table 3.

**Table 3.** Teachers' Opinions on the Difficulties of Teaching Earth Sciences.

| Teaching Difficulties | Strongly Disagree | Somewhat Disagree | Somewhat Agree | Strongly Agree | No Answer |
|---|---|---|---|---|---|
| The relationship that discipline has with time | 16.4% | 15.6% | 23.8% | 44.3% | 0% |
| The limited abstraction capacity of learners | 9.8% | 1.6% | 29.5% | 59% | 0% |
| The content of geological courses is complex and very abstract | 13.9% | 18% | 35.2% | 25.4% | 7.4% |
| The confusion of specialized vocabulary | 23% | 38.5% | 26.2 | 8.2% | 4.1% |

**Table 3.** *Cont.*

| Teaching Difficulties | Strongly Disagree | Somewhat Disagree | Somewhat Agree | Strongly Agree | No Answer |
|---|---|---|---|---|---|
| The lack of the necessary prerequisites to start the new concepts | 7.4% | 6.6% | 40.2% | 38.5% | 7.4% |
| Non-motivating knowledge for the learner | 1.6% | 23.8% | 23.8% | 47.5% | 3.3% |
| Lack of initial and ongoing training. | 9% | 25.4% | 36.1% | 27.9% | 1.6% |
| Lack of interest in the mineral environment | 12.3% | 20.5% | 26.2% | 36.9% | 4.1% |
| Social obstacles (family and society conception) | 8.2% | 23.8% | 19.7% | 45.1% | 3.3% |
| Learners do not like this discipline | 10.7% | 24.6% | 33.6% | 23.8% | 7.4% |

According to the results, the majority of the respondents to the questionnaire considered the learners' limited abstraction capacity to be the main difficulty that impeded their teaching (more than 88.5% somewhat agree or strongly agree). Another challenge was that the syllabi were not adapted to the cognitive level of the learners. This observation was confirmed during the interviews by specifying the example of plate tectonics programming and the geological phenomena associated with them for the second-year students in secondary school (14–15 years old). Furthermore, 78.7% of the questioned teachers said that the lack of prerequisites to start new geological concepts also influenced the teaching of ES. In addition, the demotivating geological knowledge taught to learners (70.3% of the teachers surveyed) mentioned that the totality of the programmed geological contents are purely theoretical, without overlooking the relationship that geology has with time and space (67.1% of teachers), and 64% of teachers mentioned the lack of initial and continuous teacher training, especially for those specializing in life sciences.

Also, 60.6% of the teachers believed that the content of the geological courses programmed in the secondary level are complex and highly abstract, and they confirmed in the interviews that the geological knowledge is not concrete or practical, and it is not linked to the learners' concerns nor to the economic concerns. This explains the learners' disaffection with ES and directly impacted the quality of learning during geology sessions. Finally, 63.1% of teachers insist on the lack of interest in the mineral environment.

As for the relationship between ES and the learner, 57.4% of respondents agreed that learners are not interested in this discipline, and less than 40% of teachers in our sample considered the ambiguity of the specialized vocabulary as a difficulty that could influence the teaching of ES in Morocco.

In addition to these difficulties, the teachers reported other difficulties influencing the quality of teaching geology in secondary school. During the semistructured interviews, they confirmed that they do not organize field trips. Geological field trips are triggering situations that allow learners to have direct contact with the environment, to ask questions, and to make hypotheses. They also have a positive impact on the relations between teachers and their learners [15] and the fight against the obstacles related to scales (of space and time) in ES [16]. Teachers who were questioned during the interviews argued that the lack of organization of geological field visits is often linked to the large number of learners in classes, the complexity of administrative procedures, time constraints, the inaccessibility of land and geological sites, and the lack of initial and continuous teacher training in this field.

It is worth mentioning that most teachers had basic training in life sciences, and these teachers are not interested in ES and consider its teaching a difficult task, especially in the absence of continuous training. Other teachers of LES connect the difficulties of teaching ES to overcrowded classrooms, which hinder the implementation of practical work and manipulations in class, as it is difficult or almost impossible to conduct experiments in their real environments. The teacher can only proceed with modelling by making mini-models on very precise time scales. This requires more caution when

interpreting the results of experiments while emphasizing the spatial and temporal conditions that are different from reality. Indeed, the modelling of reality by learners seems to be a complicated task. In addition, the laboratories of LES suffer from a lack of the necessary didactic and/or pedagogical resources for the success of the geology class, especially those related to the geology of the neighbouring regions of the schools.

Another point was raised by the teachers during the interviews about the syllabus's overload, which does not match the learners' ability to assimilate, especially qualification levels that require finishing the whole syllabus. Furthermore, the syllabus addresses many internal and external geological phenomena in only two units with new concepts.

Teachers also mentioned a point about the language of teaching in Arabic at the secondary level and French at universities, which creates a difficult transition between these two cycles for learners.

Some teachers talked about the late integration of geology in teaching sciences, although students can be initialized to it at the primary level with simple notions, as was the case in the old syllabi before the reform of 1999, but it has been reintegrated into the primary curriculum since the program review in 2018.

### 3.2. Tools Used by Teachers in Earth Science Teaching

Our survey showed that teachers use a variety of didactic tools. The majority of teachers adopt textbooks and boards containing documents from different sources (internet, university courses, specialized geology books). These tools pose a lot of didactic problems, especially when it comes to complex geological phenomena that take place on large temporal and spatial scales.

Of the surveyed teachers, 39.5% said they used information and communication techniques (ICT; video projector, digital resources, internet) in teaching units related to ES. However, the use of ICT remains limited in the teaching of LES, and the unavailability of equipment and the lack of initial and continuous training are major factors that hinder the integration of ICT in the teaching of LES as has been shown in the work [17–20]. Also the diversity of teaching aids used in a science course are indicators of effective learning [21].

Less than 30% of responding teachers reported using models, rock or thin sections, and topographic or geological maps. The non-use of these tools by other teachers in our sample is mainly explained by their absence from LES laboratories and the teachers' inability to use certain geological tools, as they did not benefit from any initial or continuing training in this area. Some teachers link it to the large number of learners in classes, the overload of syllabi (especially for the qualification levels), and the approaches adopted by the educational system.

The didactic and/or pedagogical resources used by most LES teachers in teaching ES are diverse, but their unavailability in LES laboratories and the lack of training teachers with difficulties in using certain materials are crucial issues.

### 3.3. The Usefulness of ES for a Learner According to Their LES Teacher

The answers obtained concerning the usefulness of teaching SE were treated statistically in a separate way for each answer and we have presented the data in histogram for each type of answer.

As shown in Figure 1, we found that 82% of teachers gave credit to ES in understanding the functioning of the planet. For 77% of teachers, these sciences are also essential for environmental teaching and as a source of general culture.

Furthermore, 51% of teachers attributed large importance to ES in training qualified ES teachers, and 43% believed that its function is to train engineers in mineral engineering and hydrogeology; 36% of teachers considered them essential for the training of competent geologists, and only 32% of the respondents considered that this discipline could develop a scientific spirit in the learners.

These results show that most of the teachers in our sample support the teaching of ES in Morocco. This finding corroborates the results obtained from previous surveys [11].

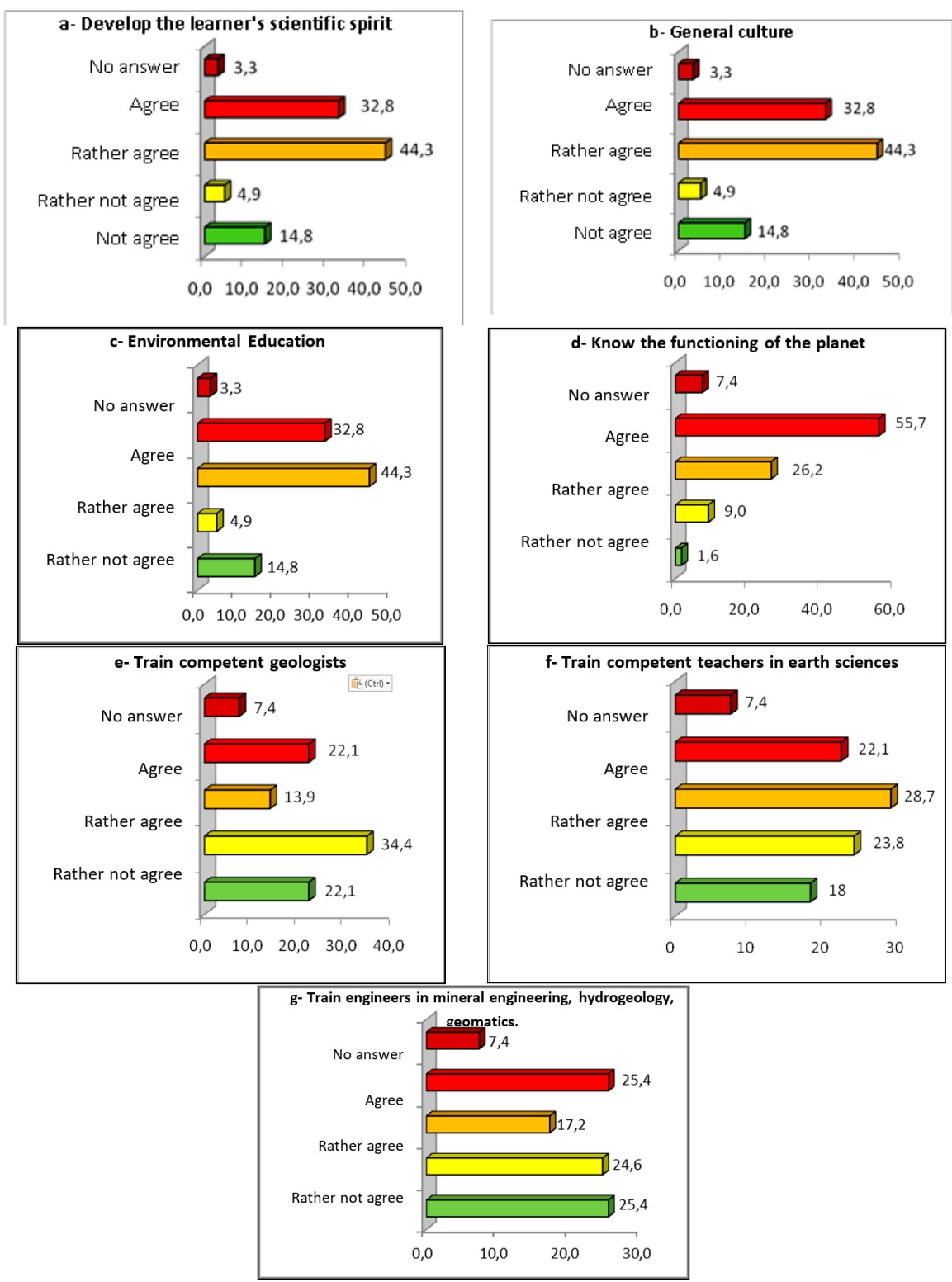

**Figure 1.** Percentages of teachers related to the usefulness of teaching Earth Sciences.

### 3.4. LES Teachers' Responses to Learners Who Are Indifferent to ES

When we asked teachers to describe their reactions to learners who are indifferent to ES, especially those who mention the useless aspect of ES learning in geology sessions, only 4.9% of teachers argued that they would motivate learners by showing them the scientific and economic benefits of ES. Another 4.1% of the surveyed teachers said they would schedule one or more sessions to discuss the scientific

and economic benefits of these sciences, and 46.7% of teachers confirmed that they would agree with the learners' opinions, explaining to them that they are obliged to teach these useless chapters and units (Figure 2).

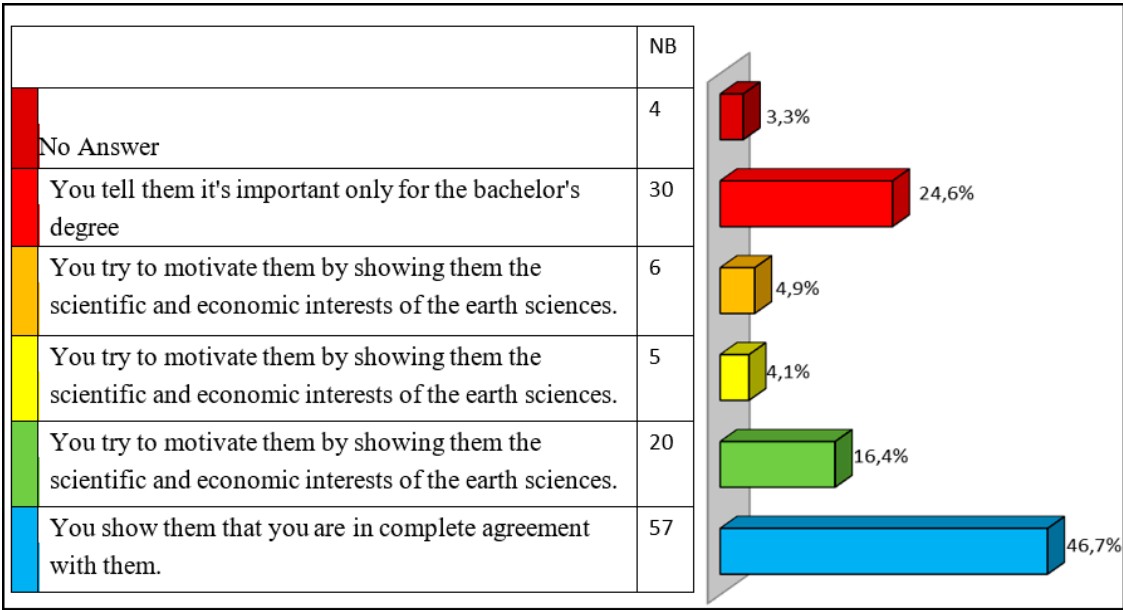

**Figure 2.** LES teachers' reaction to learner's indifference to Earth Sciences.

We then deduced that a LES teacher who teaches geology courses has limited ideas about the applied side of ES, and therefore the learners could not have enough information on ES study options. According to teachers' statements, the majority of their students choose life or economic sciences at university, and students rarely pursue their studies in ES, bearing in mind that all Moroccan universities have independent and well-developed ES departments in their science faculties, each of which has a significant number of laboratories with different specialties, fundamental or practical. Moroccan vocational guidance is limited to professions related to life sciences and neglects those related to ES.

## 4. Discussion and Conclusions

The present study was limited to highlighting some of the obstacles faced by Moroccan LES teachers in teaching and learning ES by exploring the views of teachers about this field.

Our methodology was based on a questionnaire and semistructured interviews with teachers of LES in secondary and qualifying schools who are spread acrossall the regional academies in Morocco.

The results of this study show that the teaching of ES in Morocco faces various difficulties and obstacles that could have an impact on the quality of learning in this discipline. The surveyed teachers suggested that the major difficulties that hinder the teaching of ES are related mainly to the limited ability of students to think abstractly and the lack of prerequisites in geology. Also, the teaching of this discipline was late and was not taught at all the secondary school levels. Thus, the challenges include the nonmotivational geological content taught for the learner, the relationship that geology has with time and space, and the lack of initial and continuous training of teachers, especially for those who were specialized in life sciences.

Our study has revealed other difficulties, chiefly administrative constraints, due to the insufficiency and the nonorganization of geological field trips. Similar challenges include the constraints related to the carrying out of experiments and manipulations in geology class, the low use of ICT, risk impeding the teaching of new abstract geological concepts, and consequently hindering the engagement of the learners in university courses related to ES.

The present study and analysis allowed us to diagnose the problems faced by ES teaching. Later, as science didactics, this diagnosis enabled us to suggest several ways of remediation and improvement. Curriculum and textbook redesign, revisions, and structural adjustments of natural science curriculum and textbooks would be required to adapt to learners' profiles, language levels, prerequisites, and cognitive and psychological characteristics, and facilitating the introduction of units and chapters addressing applied sciences of Earth such as geophysics, hydrogeology, and metallogeny. There is also a need to devote reasonable teaching hours as well as logical and appropriate distributions of the different chapters of the ES. Also, we should think about textbook designs related to ES specifically (as is the case of the disciplines of history–geography and physics–chemistry, each of which is taught by a single teacher, but it has two independent textbooks) in order to give the opportunity to those who are specialized in ES and its didactics to suggest syllabi and to design textbooks that suitably meet the aspirations and ambitions of the different actors in the educational community [12]. Especially our Moroccan universities possess independent and well-developed departments of Earth and universe sciences in their science faculties, and each department has a large number of laboratories with different specialties, fundamental or applied. We appreciate the curricula management decision to think about reintegrating some topics of geology into primary curricula as it was recommendedto review the decision to eliminate ES from primary school, thinking of the adoption, at an early stage at primary school levels, in a vulgarized and enjoyable way, of geological phenomena such as volcanoes [22].

From the teacher's opinions obtained in this study, we proposed the following suggestions concerning:

### 4.1. The Motivation of Learners

We hope that students should think about a real problematization of the geological knowledge so that Earth science classes become moments of scientific spirit development by motivating and engaging the learners in a process of scientific investigation, which will make them conscious actors of this approach and make them responsible for the work they are required to do.

### 4.2. Initial and Continuous Training of Teachers

For educators, teacher training is a crucial issue. Indeed, the utilization of the results and the benefits of the didactic research in sciences comes from the teachers, more precisely by their training (Martinand 1994). The teacher must be aware of the difficulties faced by teaching ES and must benefit from continuous training in ES and its didactics to overcome these obstacles, especially for those with initial training in life science. The training content should be based mainly on the practical and applied aspects, by organizing geological field trips and implementing practical work so that the teachers can master the geological concepts and phenomena, which will result in creating the necessary conceptual changes.

### 4.3. The Importance of Teaching Aids and the Use of ICT in Teaching ES

ES are characterized by geological phenomena whose space parameters, time, and physicochemical conditions we do not control as in the case of convection currents, metamorphism, plate tectonics, orogenesis, and magmatism, which require the use of analogue models or simulations. In this case, information and communication technologies (ICT) can constitute a relevant didactic tool to address complex concepts and phenomena that are difficult to acquire. For example, the possibilities offered by ICTs in certain learning situations include: Watching in reduced or accelerated model events that are not accessible to direct observation for reasons of time (e.g., very slow speeds of some geological phenomena) or space (gigantic or microscopic dimensions of certain geological objects). The moroccan study have shown that the use of simulations can improuve learning outcomes in fact that the integration of a computer simulation in a learning situation concerning relative chronology had a positive effect on student achievement [23].

Finally, it seems necessary to complete and continue this research by expanding all aspects of improvement and alternative solutions to overcome the difficulties and obstacles related to ES teaching and learning.

**Supplementary Materials:** The following are available online at http://www.mdpi.com/2227-7102/9/3/243/s1, anonymous questionnaire to secondary school teachers of Life and Earth Sciences (LES); semistructured interviews.

**Author Contributions:** Conceptualization, R.C., A.A., and S.S.; Methodology, R.C. and S.S.; Formal analysis, R.C., A.A., S.S., A.E., M.Z., and Y.B.; Investigation, R.C. and S.S.; Writing—original draft preparation, R.C.; Writing, review & editing, R.C., A.A., S.S., and M.Z.; Supervision, A.A. and S.S.; Project administration, A.A. and M.Z.; funding acquisition, R.C., A.A., S.S., and M.Z.

**Funding:** This research received no external funding.

**Conflicts of Interest:** The authors declare no conflict of interest.

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
