# Peer review of "Earth Sciences Teaching Difficulties in Secondary School: A Teacher’s Point of View"

_education, doi:10.3390/educsci9030243_

Round 1

Reviewer 1 Report

Dear Authors,

the paper is about teaching Earth Sciences and the difficulties involved according to the teachers' point of views. In general, I have no doubt that the results help in improving teaching.

I suggest some editing concerning the content and in addition if you can have an native English speaker to review the paper that would improve the quality of the text.   

Check how to use references to use them in a same way throughout the paper.

Check the literature list and correct it throughout and use one way to write the references with English translations of the titles of the paper.

Check each title and subtitle; form must be similar in every case.  

Introduction

define the Earth Science precisely by pointing out  are the natural sciences such as geology, biology etc. part of ES several 'didactics' could be said by  'research shows ... ' line 60; delete all '...INRP, Lyon FRANCE', and use scientific way of writing: the educational technology team (2014) showed that line 77; 'teachers have experienced disaffection..,' seems to be important information for to give background for this paper, but i its current form the meaning is unclear. What does that mean?use the same concept 'view' if you use it in your research question 

Methods

Tell how many teachers about altogether were in the potential target group The information about how the interview data was analysed and how much data you had etc. Describe that

Results

sample and all the 3.1.1.-3.1.3 are part of the methods; background information start the results by giving results to the research questions start each chapter with text and telling about results not by table or figure 

Conclusion

It is difficult to understand what is the point of the paragraph 'The motivation of learners', how it is linked to the results or literature: The statement 'Students should think...' is not like scientific text but more like an opinion-----------------------------------------------------Finally, I hope you find my suggestions constructive and useful to improve your paper.Best regards, Reviewer xx

Author Response

Dear reviewer

Firslyt, we would like to thank you for the time and effort to review this paper. Thank for your helpful comments and suggestions which significantly contributed to improving the quality of the work. Then, please find attached the finalized paper as well as the correction report which we have taken into consideration all  comments point by point. please find all editing corrections made in color.

Best regards

Anouar Alami

Reviewer 2 Report

This article is interesting. However, I recommend that the authors clarify the following points:

Authors should do a more comprehensive literature review regarding difficulties in acquiring ES knowledge (pages 1 and 2)

A paragraph should be added before Table 1 (page 3) summarizing the data presented.

A paragraph should be added before Table 2 (page 3) summarizing the data presented.

A paragraph should be added before Table 3 (page 4) summarizing the data presented.

Authors should develop why the tools used by teachers on Earth Science Teaching caused them a lot of didactic problems? What kind of problems? If you do not have more information on this question, you should talk about it in a section on the limits of the present study.

The methodological section needs more information related to the validation of the questionnaire carried out by the experts consulted.

A paragraph should be added before the histograms [Section: 3.4 - page 6] summarizing the data presented. The same thing on page 7.

Author Response

(The authors gave the same response as above.)
